# The Cost and the Value of Stroke Care in Greece: Results from the SUN4P Study

**DOI:** 10.3390/healthcare11182545

**Published:** 2023-09-14

**Authors:** Olga Siskou, Petros Galanis, Olympia Konstantakopoulou, Panagiotis Stafylas, Iliana Karagkouni, Evangelos Tsampalas, Dafni Garefou, Helen Alexopoulou, Anastasia Gamvroula, Maria Lypiridou, Ioannis Kalliontzakis, Anastasia Fragkoulaki, Aspasia Kouridaki, Argyro Tountopoulou, Ioanna Kouzi, Sofia Vassilopoulou, Efstathios Manios, Georgios Mavraganis, Anastasia Vemmou, Efstathia Karagkiozi, Christos Savopoulos, Gregorios Dimas, Athina Myrou, Haralampos Milionis, Georgios Siopis, Hara Evaggelou, Athanasios Protogerou, Stamatina Samara, Asteria Karapiperi, Nikolaos Kakaletsis, George Papastefanatos, Stefanos Papastefanatos, Panayota Sourtzi, George Ntaios, Konstantinos Vemmos, Eleni Korompoki, Daphne Kaitelidou

**Affiliations:** 1Center for Health Services Management and Evaluation, Department of Nursing National and Kapodistrian, University of Athens, 115 27 Athens, Greecedkaitelid@nurs.uoa.gr (D.K.); 2Department of Tourism Studies University of Piraeus, 185 34 Piraeus, Greece; 3HealThink, 570 01 Thessaloniki, Greece; 4Department of Neurology, Panarkadikon General Hospital, 221 00 Tripoli, Greece; 5Department of Neurology, General Hospital of Chania, 733 00 Creta, Greece; 61st Department of Neurology, Eginition Hospital, National and Kapodistrian University of Athens, 115 28 Athens, Greece; tountop@eginitio.uoa.gr (A.T.);; 7Department of Clinical Therapeutics, Alexandra Hospital, National and Kapodistrian University of Athens, 115 28 Athens, Greecee.korompoki@imperial.ac.uk (E.K.); 8Department of Internal Medicine, Faculty of Medicine, School of Health Sciences, University of Thessaly, 413 34 Larissa, Greecegntaios@med.uth.gr (G.N.); 91st Medical Propedeutic Department of Internal Medicine, Aristotle University of Thessaloniki, AHEPA Hospital, 546 36 Thessaloniki, Greece; 10Department of Internal Medicine, School of Medicine, University of Ioannina, 455 00 Ioannina, Greece; 11Cardiovascular Prevention & Research Unit, Laiko General Hospital of Athens at the Medical School, National & Kapodistrian University of Athens, 115 27 Athens, Greece; 12Second Department of Internal Medicine, Aristotle University of Thessaloniki, Hippokrateion General Hospital of Thessaloniki, 546 42 Thessaloniki, Greece; 13Information Management Systems Institute, ATHENA Research Center, 151 25 Athens, Greece; 14Hellenic Stroke Organization, 115 28 Athens, Greece; info@cardioresearch.net

**Keywords:** stroke, cost, burden, direct healthcare cost, loss of productivity, QALY

## Abstract

The aim of this study was to measure the one-year total cost of strokes and to investigate the value of stroke care, defined as cost per QALY. The study population included 892 patients with first-ever acute strokes, hemorrhagic strokes, and ischemic strokes, (ICD-10 codes: I61, I63, and I64) admitted within 48 h of symptoms onset to nine public hospitals located in six cities. We conducted a bottom-up cost analysis from the societal point of view. All cost components including direct medical costs, productivity losses due to morbidity and mortality, and informal care costs were considered. We used an annual time horizon, including all costs for 2021, irrespective of the time of disease onset. The average cost (direct and indirect) was extrapolated in order to estimate the national annual burden associated with stroke. We estimated the total cost of stroke in Greece at EUR 343.1 mil. a year in 2021, (EUR 10,722/patient or EUR 23,308 per QALY). Out of EUR 343.1 mil., 53.3% (EUR 182.9 mil.) consisted of direct healthcare costs, representing 1.1% of current health expenditure in 2021. Overall, productivity losses were calculated at EUR 160.2 mil. The mean productivity losses were estimated to be 116 work days with 55.1 days lost due to premature retirement and absenteeism from work, 18.5 days lost due to mortality, and 42.4 days lost due to informal caregiving by family members. This study highlights the burden of stroke and underlines the need for stakeholders and policymakers to re-organize stroke care and promote interventions that have been proven cost-effective.

## 1. Introduction

Although strokes have been documented for approximately three millennia, they remain one of the major concerns today [1]. They are the second leading cause of death, the first cause of adult-acquired disability, and the second most frequent cause of dementia worldwide. Moreover, the lifetime risk of stroke from the age of 25 years onward is almost 25% among both genders [2]. According to the World Stroke Organization’s Global Stroke Fact Sheet for 2022, in 2019, 101 million people worldwide were living with stroke, a figure that had nearly doubled over the last 30 years. Additionally, 12.2 million new strokes are diagnosed each year globally [1,3,4]. In Europe, the prevalence of stroke was 9.5 million people in 2017. In the same year, almost 1.12 million new stroke cases were diagnosed with half a million deaths attributed to a stroke [5]

Due to improvements in awareness and effective management of stroke, approximately 80% of patients survive [6,7]. Among survivors, nearly 50% live with long-term disabilities and consequently, it significantly affects their quality of life [7,8]. Stroke patients require immediate emergency and acute inpatient care, rehabilitation, home care, and outpatient pharmaceutical and medical services, all of which contribute to significant direct health expenditures. Moreover, as stroke is a long-lasting (and sometimes lifelong) condition, it is associated with increased productivity losses including work loss due to deaths and premature retirement, caregiver burden, and reduced productivity due to disease [6]. Consequently, it has a major economic impact on the health systems, communities, and families, representing one of the largest public health challenges globally. Notably, stroke is no longer considered a disease of the elderly, as over 58% of ischemic strokes occur in individuals younger than 70 years old each year, further increasing its societal impact [1].

The total burden of stroke in 2017 worldwide was estimated at USD 891 billion [1]. According to Fernandez et al. [8], the overall cost of stroke in 32 European countries in 2017 was estimated at EUR 60 billion. It was projected that the overall cost of stroke may increase to EUR 75 billion by 2030, EUR 80 billion by 2035, and EUR 86 billion by 2040. Out of EUR 60 billion, 45% was attributed to direct medical expenditure (EUR 27 billion), 27% was spent on informal, unpaid care (EUR 16 billion), 20% was related to loss of productivity (among individuals of working age) due to deaths and disability (EUR 12 billion), and 8% was attributed to social care (nursing or residential care) (EUR 5 billion).

The latest epidemiological studies for Greece show a high incidence of strokes, ranging widely from 117 to 534 new cases, with a 28-day mortality rate exceeding 20%. Moreover, the implementation of new revascularization treatments for ischemic strokes remains at low levels compared to other European countries (4.24% for thrombolysis and 0.35% for mechanical thrombectomy) [9].

Using survey data, the total burden of stroke for Greece in 2017 was estimated at EUR 650 million, of which EUR 284 million (43.7%) was allocated to direct medical costs, while the remaining costs were attributed to loss of productivity, informal/unpaid, and social care [8]. Several Greek studies have calculated the direct medical cost of stroke, primarily focusing on inpatient acute care [10,11,12]. However, no study was found that comprehensively calculated the burden of stroke throughout the care cycle using real-world data in Greece. Therefore, the motivation of the current study was to inform policy makers about the economic consequences of stroke, using real-world data.

Thus, the primary aim of this study was to measure the one-year cost of stroke in Greece (including direct medical costs, loss of productivity, and informal care costs) using a bottom-up approach with real world-data and secondary to calculate the cost per Quality-Adjusted Life Years (QALYs).

## 2. Materials and Methods

### 2.1. Analysis Framework and Study Population

We conducted a bottom-up cost analysis from the societal perspective [13], gathering resource utilization data on patients’ levels so that our calculations reflect actual cost consumption [14].

All cost components over the care cycle, including direct medical costs, productivity losses due to morbidity and mortality, and informal care costs were considered. We used an annual time horizon, encompassing all costs for the year 2021, regardless of the time of the disease onset. As far as remarkable price changes were not reported during the years of the study, we consider that this assumption has no significant impact on our results.

The study population (n = 892) was derived from the “Improving Stroke Care in Greece in Terms of Management, Costs and Health Outcomes—SUN4Patients” project, registered in ClinicalTrials.gov (NCT04109612). The SUN4P study was a prospective cohort multicenter study of patients with first-ever acute strokes, both hemorrhagic and ischemic (ICD-10 codes: I61, I63, and I64). These patients were admitted within 48 h of symptoms onset to nine public hospitals, including the National Health System and University hospitals, located in six major cities of Greece. The study took place from July 2019 to November 2021 (see Appendix A). All patients were covered by social insurance.

Detailed data were recorded for each patient, from admission up to three months after discharge. This data included demographics, clinical characteristics, diagnostic investigations, medical treatment, outcomes, and resource utilization [15]. Neurological severity upon admission was assessed using the National Institute of Health Stroke Scale (ranging from 0 to 42 points). Diagnostic investigations of the patients followed the SUN4Patients protocol. The classification into major stroke categories was based on brain imaging with CT scans or MRIs. Etiological classification of patients with ischemic stroke was determined through extensive investigations using brain, vessel, and heart imaging tests. Secondary prevention of ischemic strokes, related to antithrombotic treatment, followed existing guidelines based on sub-group classification. The modified Rankin Score was used to estimate patient handicap at discharge and three and twelve months after stroke onset (ranging from 0 to 6 points). Moreover, a sub-group of the 300 first recruited patients (derived from all study centers) were followed for a year to obtain data on post-hospital health needs, health services utilization (e.g., medication, rehabilitation, and outpatient visits), productivity loss, and the need for home care, provided by both hired health givers and family members.

The mean cost (direct and indirect) was extrapolated based on country–stroke epidemiological data to estimate the national annual burden associated with stroke.

### 2.2. Direct Healthcare Costs

We considered several major cost components in our analysis: (i) in-patient care (including both hospitalization during the first stroke episode and readmissions related to stroke during the follow up period), (ii) rehabilitation care: this encompasses both institutional-based and outpatient rehabilitation services. (iii) medication costs accounted for, (iv) outpatient visits/follow up and necessary laboratory tests, and (v) paid home care. Resources consumption (volume data) was derived from the SUN4Patients Web Platform. Unit costs, such as those of Greek DRGs, were obtained from publicly available official sources including the Ministry of Health. Additionally, the National Organization for Health Care Provision—EOPYY—which covers more than 95% of the Greek population, provides health expenditure data related to initial hospitalization and potential readmissions, outpatient pharmaceutical care, medications, and rehabilitation. For out-of-pocket expenses, we collected data through interviews with patients at the one-year follow up point. Prices were assigned to resource use to estimate total direct costs.

### 2.3. Loss of Productivity and Informal Care Cost

The major components that we took into consideration were: (i) Loss of patients’ productivity due to morbidity (absenteeism from work, early retirement, loss of work) (ii) Loss of patients’ productivity due to mortality (iii) Loss of family members’ productivity due to informal caregiving. To measure the loss of productivity due to morbidity and mortality, we employed the human capital approach [12].

Absenteeism from work (due to morbidity) was assessed based on patients’ reported productivity loss at the one-year follow up point through interviews. Loss of productivity due to mortality was calculated by taking into account the age and gender-specific number of stroke-related deaths, with working years lost being determined at the age of 65, considered the typical retirement age.

To estimate the cost of informal home care, the opportunity cost approach was adopted to calculate the value of the informal caregivers’ best alternative use for the time spent caring for their loved ones, which resulted in a loss of potential income. All volume data were derived from the patients’ interviews conducted at the one-year follow up.

The total number of lost working years (either for patients and/or for informal caregivers) was adjusted for age and gender-specific employment probabilities [8]. It was then multiplied by the average or minimum (in case of retired and unemployed informal caregivers) annual income, as obtained from OECD Health Statistics database. This amounts to EUR 16,100 in 2021 for employed or potentially employed individuals and EUR 8050 for retired and unemployed informal caregivers.

### 2.4. Quality-Adjusted Life Years

The primary clinical outcome was the modified Rankin Scale (mRS, 0–6) at the point of one year follow up (R0: no symptoms, R1: no significant disability, R2: minimal disability, R3: moderate disability, R4: moderate to severe disability, R5: severe disability, and R6: death). Utility values stratified by mRS category were derived from the literature (mRS 0 = 0.88 utilities; mRS 1 = 0.74 utilities; mRS 2 = 0.51 utilities; mRS 3 = 0.23 utilities; mRS 4 = −0.16 utilities; mRS 5 = −0.48 utilities; mRS 6 = 0 utilities) [16,17]. QALYs were calculated by multiplying the days of life (in the first 12 months) by the aforementioned utility scores.

### 2.5. Statistical Analysis

Categorical variables (e.g., gender and type of department where stroke patients are hospitalized) are presented as numbers (N) and percentages (%), while continuous variables (e.g., age and cost) are presented as mean and standard deviation (SD). The Kolmogorov–Smirnov test was applied to test the normality of the distribution of the continuous variables. Student’s *t*-test, ANOVA test, Mann–Whitney test, Kruskal–Wallis test, and Pearson’s and Spearman’s correlation coefficients were used to identify differences between variables. All tests of statistical significance were two-tailed, and *p*-values < 0.05 were considered as statistically significant. Statistical analysis was conducted with the IBM SPSS 21.0.

### 2.6. Ethics

The study protocol received approval from the Bioethics Committee of the National and Kapodistrian University of Athens Nursing Department (protocol code 277/14.01.2019) as well as the Scientific Committees of the selected hospitals where the study was conducted. Individuals were provided with both verbal and written information about the survey’s purposes, and their informed consent was obtained before enrolling them in the study. All patient data were kept strictly confidential in line with Data Protection Guidelines. Analysis was performed on anonymized data. The SUN4P design adhered to the European General Data Protection Regulation (GDPR) and was aligned with the principles set forth in Declaration of Helsinki.

## 3. Results

The study population included 892 patients, 45% of whom were admitted in non-urban hospitals and 55% in urban hospitals. The vast majority of patients (84.6%) had an ischemic stroke and they were older than 75 years on average. Details of the patients’ characteristics are presented in Table 1.

### 3.1. Total Costs of Illness and Outcomes

Based on the 32,000 cases of stroke among Greeks in 2021, the short-term (one-year) burden of illness was estimated at EUR 343.1 mil. (EUR 10,722/patient). This is equivalent to a mean one-year cost of EUR 23,308 per QALY.

Out of the EUR 343.1 million for the total burden of strokes, more than 53% was attributed to direct health care costs while almost 47% was related to loss of productivity. Detailed data about costs and QALYs per type of stroke are presented in Table 2.

The vast majority of direct healthcare costs consisted of public expenditure (88%, i.e., EUR 162 million) funded mainly by EOPYY and the Ministry of Health covering operational expenses and NHS personnel wages, respectively. The remaining 12% of total health care costs (i.e., about €21 million) consisted of out-of-pocket payments.

### 3.2. Average Direct Healthcare Cost per Type of Care and Type of Stroke

In order to interpret cost results and to bring into light the potential gaps to optimal healthcare resulting in restriction of the value of care, some core components of healthcare are indicatively presented in Table 3, along with outcomes at the point of discharge. The remaining healthcare utilization cost components have been analyzed as well and presented in detail in a recently submitted manuscript for peer review and publication [18].

Average direct healthcare costs per type of care and type of stroke are presented in Figure 1. Our analysis indicated that both for hemorrhagic and ischemic stroke, inpatient care was the cost driver. Hemorrhagic stroke’s average direct healthcare cost was found by 32% higher when compared to ischemic stroke (average healthcare cost) (*p* < 0.05).

In relation to home care, based on the results of the interviews conducted at the point of the one-year follow-up (n = 324), about 7% of stroke survivors stated being in need of a paid caregiver, for a period of 6.3 months after discharge with an average monthly wage of EUR 544 (SD = 106). Of note, the vast majority of paid caregivers were immigrants (83.3%) without any training in stroke patients’ home care. It is worth mentioning that costs related to home care were entirely funded by out-of-pocket payments (EUR 6.1 million in total).

### 3.3. Loss of Productivity

The average productivity losses among stroke patients over the course of one year were estimated to be 116 work days. This included 55.1 days lost due to premature retirement and absenteeism from work, 18.5 days lost due to mortality, and 42.4 days lost due to informal caregiving by family members.

In total, productivity losses among stroke patients (n = 32,000) were calculated at EUR 160.2 million. This figure included an estimated EUR 110.5 million (69% of indirect cost) attributed to productivity losses due to premature mortality and/or absence from work (early retirement or absenteeism) and EUR 49.7 mil (31% of indirect cost) related to informal/unpaid care costs (Table 2).

Notably, almost 25% of survivors at the point of the one-year follow up reported receiving support from an informal caregiver who had an average age of 60.32 years (SD = 12.4). Patients’ wives comprised the majority (68%) of informal caregivers, with 80% being married and over half (57%) being professionally active.

## 4. Discussion

We conducted a bottom-up cost analysis from a societal perspective to assess the burden of stroke over a one-year cycle. Τhe analysis was based on real-world data collected prospectively from nine public and university hospitals in various Greek cities as part of the SUN4P project. Additionally, healthcare resource costs for participants were directly provided by the National Organization for the Provision of Health Care (EOPYY), following an agreement with EOPYY.

To the best of our knowledge, this study represents the first cost-of-illness analysis related to stroke management in Greece over a 12-month period, based on real-world data that was prospectively collected, with healthcare cost data directly supplied by EOPYY. Another noteworthy strength of our study is that data on the loss of productivity due to morbidity and informal caregiving were obtained directly from patients or their relatives through interviews conducted at the one-year follow up, in accordance with study protocol.

Our findings indicate that the total cost of stroke in Greece for the year 2021 amounted to EUR 343.1 million. Of this total, EUR 182.9 million (53.3%) was attributed to direct healthcare costs, representing 1.1% of the current health expenditure in 2021. This places Greece in the middle range among European countries, where respective rates range from 0.58% to 4.34% [8]. Furthermore, our study revealed that 46.7% of the economic burden of stroke (EUR 160.22 mil.) referred to non-health areas (indirect cost), a figure in line with the corresponding mean rate observed in European countries (47%) [8].

There is limited evidence from Greece to compare the total cost of stroke (in monetary units) estimated in our study in a population-based cost analysis study, conducted to assess the overall health and social costs of stroke in 32 European countries, including Greece [8]. The estimated overall cost of stroke in Greece was higher in 2017 when compared to our results. This difference can be attributed to variations in the study design. The Fernandez et al. study [8] utilized a top-down approach [8]. It gathered information on self-reported stroke patients (ICD-10 codes I60-I69: approximately 34,000 individuals) and their resource utilization across primary care, outpatient care, emergency services, social services, and informal caregiving through surveys (e.g., SHARE database and Health Interview Survey 2014), while inpatient care data were retrieved from the Eurostat database. In contrast, our study employed a bottom-up cost analysis taking into consideration stroke patients (ICD-10 codes: I61, I63, and I64: n = 32,000 in total at the country level), diagnosed by specialized physicians, resources used, and costs based on the SUN4Patients registry and third-party payroll real-world data (EOPYY). In addition to this, in the case of the Fernandez et al. study [8], the friction method was applied to calculate the loss of productivity, considering only the time required to replace a worker with another from the pool of the unemployed. Taking into account EUR 24,800 yearly earnings for men and EUR 20,500 for women in 2017. In our analysis, we adopted the human capital approach [13] to measure loss of productivity. We considered average annual wages, which amounted to EUR 16,100 in 2021 for employed or potentially employed individuals and EUR 8050 for retired and unemployed informal caregivers, based on data from the OECD Health Statistics.

With respect to the allocation of direct healthcare costs, our findings align with previous European research [19], which also identified inpatient care as the primary cost driver. In addition, our study revealed that the average cost of hemorrhagic stroke was higher when compared to ischemic stroke cost. This disparity can be attributed to the heightened healthcare needs of hemorrhagic stroke patients, necessitating longer-term and more intensive healthcare services [20]. For instance, we observed that the median length of hospital stay for hemorrhagic stroke patients was 50% longer than that of ischemic stroke patients resulting, consequently, in increased cost of hospitalization. However, in outpatient pharmaceutical care, our results reveal that ischemic stroke survivors were in need of increased pharmaceutical care (in order to improve risk factors control, e.g., blood pressure, blood glucose, lipid profile, etc.) for the secondary prevention of stroke recurrence [21], resulting to increased mean (out-patient) pharmaceutical care cost.

Regarding non-healthcare costs, prior studies have extensively documented the impact of stroke on both patients and their relatives or informal caregivers’ productivity losses [8,12,19]. Our analysis corroborated these findings, revealing significant productivity loss within the first year following a stroke. Average productivity losses among stroke patients (over the cycle of one year) were estimated at 116 work days. Of note, significant consequences on stroke patients’ families were found, in alignment with previous international and national literature [22,23] underlying that “caring for a loved one affected by stroke puts a significant burden on the family caregiver” [22]. Our calculations further demonstrated that nearly one-third of the total productivity loss comprised of informal (unpaid) care costs (i.e., about EUR 50 mil., representing 14.5% of total stroke burden), incurred by family members. The increased burden of family members when caring for a stroke patient could be attributed to various factors, including the scarcity of nursing homes, an insufficient number of rehabilitation centers, and limited support programs for home-based care. These factors, coupled with the Greek tradition of relying on relatives for home care of the chronically ill and disabled [23,24,25], contribute to the elevated caregiving burden.

Τo the best of our knowledge, this study represents the first attempt to quantify the value of stroke care in Greece defined as the cost per QALY over the course of one year. Our calculations yielded a cost of stroke care per QALY at EUR 23,308, a figure equivalent to 1.44 times the average annual wage in Greece in 2021, as reported by the OECD Statistics Database. Even though there is no evidence from Greece to compare these results with, one recent publication (2023) from New Zealand [26] which is similar to our study’s population baseline characteristics, provided costs, and outcomes data based on a sample of 1510 acute stroke (elderly) patients. According to their findings, the cost per QALY ranged from EUR 34,944 (in the case of patients admitted to non-urban hospitals) to EUR 38,064 (in the case of patients admitted to urban hospitals) in 2018, resulting in a figure almost equal to the average annual wage in New Zealand in 2018 based on the OECD Statistics Database (2023). Comparatively, the cost per QALY in Greece appears to be relatively higher which may be attributed in part to the lower QALYs gained within the Greek context and potential inefficiencies in the organization of stroke care. Over the short-term span of one year, the average QALYs were estimated at 0.46 (0.38) in Greece. In contrast, the study by Kim and colleagues, [26] reported a range from 0.46 (for those patients admitted to non-urban hospitals accounting for 40%) to 0.54 (for those patients admitted to urban representing 60%). These findings suggest potential gaps in the delivery of optimal stroke care within the Greek healthcare system.

Our findings underscore the challenges in stroke care within Greece. Only a minority of patients (14%) had the opportunity to be admitted to a specialized Acute Stroke Unit (ASU), due to the country’s limited availability of ASUs (0.6/million population vs. >2/million population in most European countries) [26]. Moreover, we observed relatively low rates of rtPA administration, 4.6% (in our study) vs. 7.3% the average European rate [27], that could be attributed to delays from stroke onset to the first scan, especially increased during the COVID-19 pandemic, when our study was conducted almost simultaneously. Moreover, another obstacle to the provision of rtPA administration was, in some cases, hospitals’ limited capacity [11,18,28], given the country’s insufficient number of ASUs. Our analysis also revealed gaps in rehabilitation care. Only one-third of patients participated in an early rehabilitation program during hospitalization and approximately 13% of survivors were admitted to a rehabilitation center after discharge, although 24.3% of survivors had a mRS = 4–5, at the point of discharge. Suboptimal rehabilitative care could probably be attributed to insufficient financing, as less than 1% of the Current Health Expenditure (CHE) in Greece consists of rehabilitative care, while the corresponding rate in most European countries ranges from 2% to 5% of the CHE [29].

Not surprisingly, we found that hemorrhagic stroke patients’ average cost per QALY was over doubled compared to the respective rate of ischemic stroke patients. Indeed, hemorrhagic stroke is related to worse functional and clinical outcomes [30] resulting in decreased QALYs and more costly healthcare compared to ischemic stroke [19]. Also, increased loss of productivity was reported in the case of hemorrhagic stroke patients, as our analysis revealed that out of EUR 160.2 mil., one-third was incurred by hemorrhagic stroke patients (who account for only 15% of total stroke patients). Indeed, 61.5% of the total hemorrhagic stroke burden referred to loss of productivity (vs. 41.1% in the case of ischemic stroke) due to increased fatality and severity/disability compared to ischemic stroke [20].

Our study has some limitations that have to be considered in the extrapolation of the results, as this is not a nationwide registry. Indeed, in our analysis, the study population consisted of stroke patients admitted only to public (NHS and University) hospitals, based on the SUN4Patients protocol. Thereby, our calculations related to direct (mainly out-of-pocket) healthcare expenditure are likely to be underestimated. EOPYY data indicate that approximately 12% of all stroke patients are admitted to private hospitals with a remarkably higher inpatient cost covered mainly by household budgets and/or private insurance. Patients treated in private hospitals (reflecting improved socio-economic profile) are probably willing to pay increased out-of-pocket payments to ensure faster access to improved healthcare after discharge (e.g., rehabilitation, paid home care, etc.), resulting in higher overall direct healthcare costs. In addition, a recent study aimed to measure the total annual economic burden of atrial fibrillation (AF)-related stroke in Greece [31] reported that 39% of the direct healthcare costs for stroke was financed by the patients via out-of-pocket expenses, while in our analysis the respective rate was 12%. While we acknowledge this limitation, it is important to emphasize that policy makers wield significant influence over publicly funded healthcare services. Consequently, the greatest potential for policy impact lies within this domain. This primary reason guided our decision to focus on patients admitted to public hospitals. Moreover, the carotid intervention cost was not included as they were implemented only for six patients. Finally, emergency ambulance services cost associated with stroke care in the pre-hospital period was not included in the cost analysis, due to a lack of data.

### Policy Implications

The results of our study highlight the necessity of re-organizing stroke care in Greece, with the full implementation of comprehensive continuous stroke services in order to achieve improved outcomes and thus increased value of care. Indeed, based on the results from a literature review conducted by the European Brain Council Value of Treatment, full implementation of comprehensive stroke services was related to an absolute decrease in risk of death or dependency (by 9.8%) [32].

In the case of Greece, stakeholders and governmental officials should pay attention to further increasing reperfusion therapy rates (Although significant progress has been achieved during the last years, as in 2018, just a 1% rtPA administration rate was reported for Greece in a European study, further efforts have to take place in order to reach the corresponding European rate [27,28]) and in developing/expanding specialized Acute Stoke Units —ASUs (at least 8–10 over the country), staffed with well trained and dedicated stroke teams [18]. Although significant investments are required to ensure these services, the current literature indicates their cost-effectiveness.

Previous research [33,34,35,36] indicates that reperfusion therapy constitutes a cost-saving or cost-effective treatment option compared to traditional treatment for eligible acute ischemic stroke patients. Moreover, a national cost-effectiveness analysis with data derived from the SUN4Patients registry concluded that rtPA is a dominant option for the management of eligible stroke patients from the third-party payer perspective, given that it is more effective and costs less than conservative treatment. In particular, rtPA led to 0.009 incremental QALYs per patient in the first three months in Greece, with the total cost per patient administered rtPA was estimated at EUR 2196.65, vs. EUR 2499.45 in the conservative treatment group [37].

In addition, researchers proved that admission to a specialized ASU is related to improved clinical outcomes and shorter lengths of stay, compared to conventional treatment in internal medicine or neurological departments, resulting in the cost-effectiveness of ASUs [38]. In Greece, given the country’s geographical disparities, Mobile Stroke Units (MSU) could also contribute to improving the value of care, especially in non-urban areas. In accordance with a recently published study from the Norwegian Acute Stroke Prehospital Project, acute ischemic stroke patients’ management, the use of mobile stroke units (MSUs) reduces onset-to-treatment time and increases thrombolytic rates. In addition, there is evidence that MSU settings are potentially cost-effective compared to conventional care, depending on the annual number of treated patients per MSU (the higher the number treated in MSUs, the more cost-effectiveness is achieved) [39].

Finally, gaps in rehabilitation services indicate the necessity to implement effective rehabilitation programs (e.g., timely admission to rehabilitation centers for those patients in need and inpatient rehabilitation) to improve physical functionality and quality of life and thereby reduce the need for longer-term care, resulting in cost constraints. Based on the results of our study, improved mRS at the point of discharge was related to decreased average healthcare costs over the cycle of care. Previous research has proved inpatient rehabilitation as a cost-effective intervention, especially for partially self-sufficient and moderately disabled stroke patients [40].

Moreover, taking into consideration the country’s insufficient financing of rehabilitative care due to budget constraints and increased dependency of stroke patients on their family members, which results in significant loss of productivity, alternative interventions, such as home-based rehabilitation, ought to be examined. Based on the results from a study aimed to explore the cost-effectiveness of home-based vs. center-based rehabilitation in stroke patients across 32 European countries, home-based rehabilitation was found to be highly likely to be cost-effective (>90%) in the vast majority of the European countries included in the study, a finding also confirmed in the case of Greece [41].

## 5. Conclusions

To the best of our knowledge, this study represents the first attempt in Greece to comprehensively measure the total cost of stroke and evaluate the value of care over a cycle of one year based on real-world data. These findings underscore the imperative for stakeholders and policy makers to reconsider the organization of stroke care in Greece. It emphasizes the importance of promoting interventions that have proven to be cost-effective, such as increasing the rate of thrombolysis, enhancing public funding for rehabilitation, and implementing organized programs to alleviate the burden of family members.

## Figures and Tables

**Figure 1 healthcare-11-02545-f001:**
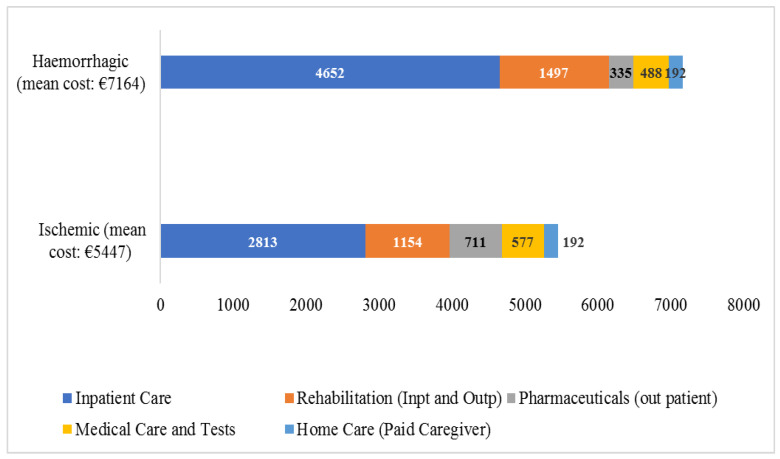
Mean direct healthcare cost (in euros) per type of care and type of stroke.

**Table 1 healthcare-11-02545-t001:** Basic study population characteristics.

	All Patients (N_All_ = 892)	Ischemic Stroke (N_Isch_ = 755)	Hemorrhagic Stroke (N_Hem_ = 137)	*p*-Value
**Age**, mean (SD)	75.6 (13.5)	75.6 (13.6)	75.8 (13.2)	0.419
**Age groups**				0.653
<65	186 (20.9)	159 (21.1	27 (19.7)	
65–74	156 (17.5)	136 (18.0)	20 (14.6)	
75–84	297 (33.3)	246 (32.6)	51 (37.2)	
85+	253 (28.4)	214 (28.3)	39 (15.4)	
**Gender** (Men)	447 (50.1)	368 (48.7)	79 (57.7)	0.063
**mRS** 0–1 prior to admission	704 (78.9)	591 (78.3)	113 (82.5)	0.306
**NIHSS** scale at admission, median (interquartile range)	7 (3–12)	6 (3–11)	12 (5–23)	<0.001
**Discharge destination** in patients or relatives’ home (781 alive at discharge)	649 (83.1)	587 (85.4)	62 (66.0)	0.000
**History of Stroke risk factors**				
Hypertension	616 (691.1)	522 (69.1)	94 (68.6)	0.920
Diabetes	230 (25.8)	208 (27.5)	22 (16.1)	0.004
Current smoking	211 (23.7)	184 (24.4)	27 (19.7)	0.275
Hyperlipidemia	328 (36.8)	288 (38.1)	40 (29.2)	0.054
Atrial Fibrillation	265 (29.7)	228 (30.2)	37 (27.0)	0.478
Coronary Artery Disease	117 (13.1)	101 (13.4)	16 (11.7)	0.680
Previous TIAs	81 (9.1)	66 (8.8)	15 (10.9)	0.419
**Classification of ischemic strokes**				
Large vessel atherosclerotic		71 (9.4)		
Cardioembolic		240 (31.8)		
Lacunar		117 (15.5)		
Other		16 (2.1)		
Cryptogenic		311 (41.2)		
1 year mortality	220 (24.7)	162 (21.5)	58 (42.3)	0.000

Values are expressed as n (%) unless otherwise indicated. mRS: modified Rankin Scale, NIHSS: National Institute of Health Stroke Scale, and TOAST classification.

**Table 2 healthcare-11-02545-t002:** Overall burden of stroke and cost per QALY during the short term of one year in Greece (n = 32,000).

Cost in Million Euros	Ischemic Stroke	Haemorrhagic Stroke	All Types of Stroke	% of Total Cost
I. Direct Healthcare Cost	147	35.9	182.9	53.3
Inpatient Care	75.9	23.3	99.2	28.9
Rehabilitation (Inpt and Outp)	31.2	7.5	38.7	11.3
Pharmaceuticals (Outp)	19.2	1.7	20.9	6.1
Medical Care and Tests	15.6	2.4	18	5.2
Home Care (paid caregiver)	5.1	1.0	6.1	1.8
II. Loss of Productivity	102.68	57.54	160.22	46.7
Loss of Productivity due to Morbidity (premature retirement and absenteeism from work)	59.455	23.266	82.72	24.1
Loss of Productivity due to Mortality	3.486	24.286	27.772	8.1
Informal Care costs	39.742	9.985	49.727	14.5
Total = I + II	249.66	93.44	343.1	100
QALYs				
Total QALYs	12,555.6	2110.2	14,720	
Mean QALYs (SD)	0.4638 (0.4616)	0.4282 (0.4138)	0.46 (0.38)	
Total Cost/QALY (in euros)	19,884	44,281	23,308	*p* < 0.005

**Table 3 healthcare-11-02545-t003:** Core components of the provision of healthcare and outcomes at the point of discharge.

	All Patients (N_All_ = 892)	Ischemic Stroke (N_Isch_ = 755)	Hemorrhagic Stroke (N_Hem_ = 137)	*p*-Value
ALoS-Average Length of Stay, median (interquartile range)	6 (4–10)	6 (4–9)	9 (6–15)	<0.001
Treated in Specialized ASU-Acute Stroke Unit	127 (14%)	116 (15.4%)	11 (8%)	0.024
rtPA Administration: Eligible /Undertook		109 (14.4%)/35 (4.6%)		
Early inpatient rehabilitation	271 (30.4%)	225 (29.8%)	46 (33.6%)	>0.05
Survivors’ (n = 772) admission to Rehabilitation center after discharge	105 (13.6%)	83 (12%)	22 (23.4%)	<0.001
mRS at discharge				<0.001
0–1	39%	43%	16.8%	
2–3	24.3%	25.2%	19.7%	
4–5	24.3%	22.9%	32.1%	
Dead	12.3%	8.9%	31.4%	

## Data Availability

Additional data at patient level isn’t available due to privacy (GDPR) and ethical restrictions.

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
