# Peer review of "The Cost and the Value of Stroke Care in Greece: Results from the SUN4P Study"

_healthcare, 2023, doi:10.3390/healthcare11182545_

Round 1
Reviewer 1 Report
Dear authors
Thank you for submitting your paper
There is only one aspect in my opinon must be improved or at least explained with more data
In table 2 the amount of euros lost by productivity is high. It is explained in line 225 and posterior. but at hte same time the mean age of your population is 75 years old. In my opinión a more clear explanation about the origin from the loss of productivitywill be great because old people usually are not employed, bur 31% of the indirect cost are produced by absenteeism or early retirement. The amount is elevated thinking in an old population that probably is retired previously to the stroke.
If you have more data to explain this aspect, could be great to made it clearer.
Author Response
Dear Reviewer,
Thank you so much for providing this valuable comment, helping us to improve our manuscript.
Based on your comment, we revised table 1, adding data related to the study population and destination after discharge. Indeed, almost 21% of the understudy population was <65 years old probably professionally active. Moreover, for the vast majority (>80%) of the under-study population destination after discharge was home, indicating burden on younger family members (probably professionally active)
Kind regards,
Olga Siskou
Reviewer 2 Report
The manuscript presents a study on the measurement of the one-year total cost of stroke and the evaluation of stroke care value in terms of cost per Quality-Adjusted Life Year (QALY). The study contributes valuable insights into the economic burden of stroke and underscores the necessity for strategic interventions. The methodology and findings are well-structured, but some revisions and clarifications are needed to enhance the manuscript's clarity and impact.
The manuscript should provide a more detailed explanation of the "bottom-up cost analysis from the societal point of view." Readers would benefit from a clearer description of how each cost component (direct medical costs, productivity losses, and informal care costs) was calculated and integrated into the analysis. Additionally, a more detailed breakdown of how the sample size of 892 patients was distributed cardioembolic and non-cardioembolic stroke would be helpful to understand the study population better.
While using an annual time horizon for cost analysis is reasonable, it's important to acknowledge the potential limitations of assuming that all costs occurred in the year 2021, regardless of disease onset. Some discussion on how this assumption might impact the results and potential sensitivity analyses would provide a more comprehensive perspective.
The concept of evaluating stroke care value in terms of cost per QALY is intriguing. However, the manuscript could provide additional context on how the €23,308 per QALY figure aligns with existing thresholds or standards for cost-effectiveness in healthcare. This would help readers better gauge the implications of this value and its interpretation. The tables include the cost of rtPA in case of ischemic stroke, but we receive no information about the rate and cost of imaging studies performed or that of interventional treatment. A thorough discussion on this topic supplemented with evaluation of medical treatment strategies would greatly benefit the reader, particularly in understanding the decision-making process when it comes to selecting the appropriate therapeutic strategy for different types of ischemic strokes. (see PMID: 34063551) How can the identified costs and value influence decision-making for healthcare stakeholders and policymakers? Offering actionable insights would add depth to the manuscript's conclusion.
The manuscript should maintain consistent terminology, particularly when referring to costs. At times, "costs" are mentioned as "cost components," which could lead to confusion.
Author Response
Dear Reviewer,
Thank you so much for providing these valuable comments, helping us to improve our manuscript. Please find below (in blue colour) our replies related to the way we met each comment
The manuscript presents a study on the measurement of the one-year total cost of stroke and the evaluation of stroke care value in terms of cost per Quality-Adjusted Life Year (QALY). The study contributes valuable insights into the economic burden of stroke and underscores the necessity for strategic interventions. The methodology and findings are well-structured, but some revisions and clarifications are needed to enhance the manuscript's clarity and impact.
The manuscript should provide a more detailed explanation of the "bottom-up cost analysis from the societal point of view." Readers would benefit from a clearer description of how each cost component (direct medical costs, productivity losses, and informal care costs) was calculated and integrated into the analysis. Additionally, a more detailed breakdown of how the sample size of 892 patients was distributed cardioembolic and non-cardioembolic stroke would be helpful to understand the study population better.
We added a short explanation about bottom up approach and we added a reference (lines 128-129). In sections 2.2 and 2.3 are described the calculations of each cost component
We added in Table-1 more details regarding our patients characteristics. Cardioembolic strokes is frequent (31.8% of all ischemic strokes)
While using an annual time horizon for cost analysis is reasonable, it's important to acknowledge the potential limitations of assuming that all costs occurred in the year 2021, regardless of disease onset. Some discussion on how this assumption might impact the results and potential sensitivity analyses would provide a more comprehensive perspective.
We added a sentence in the end of the first paragraph of the methodology explaining that during the years of our study no remarkable price changes were reported and therefore this assumption has no significant impact on our results
The concept of evaluating stroke care value in terms of cost per QALY is intriguing. However, the manuscript could provide additional context on how the €23,308 per QALY figure aligns with existing thresholds or standards for cost-effectiveness in healthcare. This would help readers better gauge the implications of this value and its interpretation. The tables include the cost of rtPA in case of ischemic stroke, but we receive no information about the rate and cost of imaging studies performed or that of interventional treatment. A thorough discussion on this topic supplemented with evaluation of medical treatment strategies would greatly benefit the reader, particularly in understanding the decision-making process when it comes to selecting the appropriate therapeutic strategy for different types of ischemic strokes. (see PMID: 34063551) How can the identified costs and value influence decision-making for healthcare stakeholders and policymakers? Offering actionable insights would add depth to the manuscript's conclusion.
Thanks for these comments! Indeed, evaluation of medical treatment strategies is a very “hot” interesting subject both for health professionals and policy makers. Thus, we have submitted another paper related to cost effectiveness of thrombolysis using real word data from the SUN4P study. Moreover, detailed data about the cost of diagnostic imaging etc is presented in a previously published paper (IOS Press Ebooks - Inpatient Cost of Stroke Care in Greece: Preliminary Results of the Web-Based “SUN4P” Registry)-see table 1. The cost per QALY is discussed in lines 378-401
However, as the purpose of the current study was to provide an overview of the stroke cost all over the cycle of care in Greece (using for the first time real world data), we consider, that in case we will add more details, maybe the paper will become very extensive.
Kind regards,
Olga Siskou
Reviewer 3 Report
This paper “Τhe cost and the value of stroke care in Greece: Results from the SUN4P Study” aimed to measure one-year total cost of stroke and to investigate the value of stroke care, defined as cost per QALY. The study population included 892 patients with first ever acute stroke, hemorrhagic and ischemic, (ICD-10 codes: I61, I63 and I64) admitted within 48 hours of symptoms onset to nine public hospitals located in six cities. The authors conducted a bottom-up cost analysis from the societal point of view. All cost components including direct medical costs, productivity losses due to morbidity and mortality and informal care costs were considered. The authors used an annual time horizon, including all costs for 2021 irrespective of the time of disease onset. The average cost was extrapolated to estimate the national annual burden associated with stroke. The authors estimated the total cost of stroke in Greece at €343.1 mil a year in 2021, (€10,722/patient or €23,308 per QALY). Out of €343.1 mil., 53.3% (€182.9 mil) consisted direct healthcare cost representing 1.1% of current health expenditure in 2021. Overall, productivity losses were calculated at €160.2 mil. The mean productivity losses were estimated to 116 workdays with 55.1 days lost due to premature retirement and absenteeism from work, 18.5 days lost due to mortality and 42.4 days lost due to informal caregiving by family members. Basically, this study highlights the burden of stroke and underlines the need for stakeholders and policy makers to re-organize stroke care and promote interventions that have been proved as cost-effective.
The topic is justified, and the paper is well-organized. However, the paper could be further improved if the following remarks are taken into consideration:
1. ABSTRACT: summarize this text on the abstract.
2. A few of the grammatical and flow mistakes are found in the draft.
3. Introduction is well written, but justification of the research is little bit missing.
4. Last paragraph of an introduction section may contain organization of the research article.
5. Some more statistical analysis is considered for the attributes of the population, would enhance presentation and conclusions briefly.
6. The motivation of the study is not clear.
7. This type of study is almost necessary in every region, culture, and even individual healthcare centres to assist the government in their policy making.
minor edits required
Author Response
Dear Reviewer,
Thank you so much for providing these valuable comments, helping us to improve our manuscript. Please find below our replies (in blue color) related to the way we met each comment
- ABSTRACT: summarize this text on the abstract.
Could you please explain in more details, what are you proposing to do?
- A few of the grammatical and flow mistakes are found in the draft.
OK, we have proceeded with major revisions
- Introduction is well written, but justification of the research is little bit missing.
OK, we added a few sentences by the end of the introduction (prior materials and methods)
- Last paragraph of an introduction section may contain organization of the research article.
OK, we have revised the aim of the study
- Some more statistical analysis is considered for the attributes of the population, would enhance presentation and conclusions briefly.
OK, we enriched Table 1
- The motivation of the study is not clear.
The motivation of the study is to inform policymakers and the scientific community about the effectiveness of the management and cost of stroke in Greece, using real world data. We added this sentence prior the aim (in the introduction)
- This type of study is almost necessary in every region, culture, and even individual healthcare centres to assist the government in their policy making.
OK we tried to underline these issues
Kind regards,
Olga Siskou
Reviewer 4 Report
Comments:
This is an article evaluating the cost and the value of stroke care in Greece. The research question, although not novel, but is clinically important one. It emphasizes the enormous economic burden on the family, society and the nation. The study design is appropriate. The analysis and it’s interpretations are adequate. The manuscript is well structured and easy to read. The authors need to address the following points:
1- The authors mention that One out of three patients were followed for 1 year for reporting the health care resources consumption. How these patients were selected? What factors were considered? Were any stratification of various geographic, socio-economic, and patient parameters considered, so that it may be representative of the population.
2- Did the outpatient follow up visits also included the cost borne by the accompanying care givers?
3- What was the number of mortality among the patients at 1 year?
4- The investigators have considered only the patients admitted with stroke. They have not included patients presenting to out-patient departments with minor strokes ,who might not have been admitted.
5- The ‘inpatient care, contributed highest to the health care cost. Did the investigators tried to look at different components of it..e.g icu stay, investigations, drugs, procedures etc?
6- Was any intervention (which may be elective) e,g Carotid stenting/Endarterectomy included in the cost analysis?
7- Authors mentioned about rtPA in the component of healthcare provision (Table 3). Was mechanical thrombectomy also included? How many received mechanical thrombectomy?
8- Any emergency ambulance services associated with stroke care in pre-hospital period may also be included in the cost analysis.
9- It will be interesting to know, how many of the patients had insurance coverage? A statement on presence/absence of any nationalized health insurance policy in the discussion part will be informative for the readers.
Minor editing of the English language is required
Author Response
Dear Reviewer,
Thank you so much for providing these valuable comments, helping us to improve our manuscript. Please find below our replies (in blue color) related to the way we met each comment
1. The authors mention that One out of three patients were followed for 1 year for reporting the health care resources consumption. How these patients were selected? What factors were considered? Were any stratification of various geographic, socio-economic, and patient parameters considered, so that it may be representative of the population.
OK, we added by the end of §2.1 a sentence clarifying that the first 300 patients recruited (derived from all centers) were followed up for a year.
It was impossible to follow up for one year all the patients due to time limitations
2. Did the outpatient follow up visits also included the cost borne by the accompanying care givers?
Yes, of course. We mention it in §2.2
3. What was the number of mortality among the patients at 1 year?
It was 24.7%. We added it in Table 1
4- The investigators have considered only the patients admitted with stroke. They have not included patients presenting to out-patient departments with minor strokes, who might not have been admitted.
Yes, we confirm that we didn’t include patients presenting to out -patients departments with minor strokes, based on our protocol that is described in the second paragraph of §2.1
5. The ‘inpatient care, contributed highest to the health care cost. Did the investigators tried to look at different components of it..e.g icu stay, investigations, drugs, procedures etc?
All cost components of inpatient care were taken into account, to estimate the sum of inpatient care cost. More details are reported (IOS Press Ebooks - Inpatient Cost of Stroke Care in Greece: Preliminary Results of the Web-Based “SUN4P” Registry)
6. Was any intervention (which may be elective) e,g Carotid stenting/Endarterectomy included in the cost analysis?
No, only six carotid interventions were reported. Thus, we added this exclusion to limitations
7. Authors mentioned about rtPA in the component of healthcare provision (Table 3). Was mechanical thrombectomy also included? How many received mechanical thrombectomy?
Only one patient received mechanical thrombectomy. We added in introduction data related to the limited number of mechanical thrombectomies, implemented in Greece.
8. Any emergency ambulance services associated with stroke care in pre-hospital period may also be included in the cost analysis.
No, it was not included due to lack of data. We added it by the end of the limitations.
9. It will be interesting to know, how many of the patients had insurance coverage? A statement on presence/absence of any nationalized health insurance policy in the discussion part will be informative for the readers.
All of the under-study patients were covered by social insurance. We added this data by the end of the second paragraph of §2.1
Kind regards,
Olga Siskou
Round 2
Reviewer 2 Report
Thank you for including me in the revision of the article. Important progress was made in the presentation of the data. All my comments were adequately answered.